# On the interplay of temporal resolution power and spatial suppression in their prediction of psychometric intelligence

Lisa M. Makowski[1], Thomas H. Rammsayer[1], Duje Tadin[2], Philipp Thomas[1], Stefan J. Troche[1]*

**1** Institute of Psychology, University of Bern, Bern, Switzerland, **2** Department of Brain and Cognitive Sciences, Neuroscience, Ophthalmology and Center for Visual Science, University of Rochester, Rochester, NY, United States of America

* stefan.troche@unibe.ch

**Data Availability Statement:** All relevant data are available from the BORIS database (https://doi.org/10.48620/62).

## Abstract

As a measure of the brain's temporal fine-tuning capacity, temporal resolution power (TRP) explained repeatedly a substantial amount of variance in psychometric intelligence. Recently, *spatial suppression*, referred to as the increasing difficulty in quickly perceiving motion direction as the size of the moving stimulus increases, has attracted particular attention, when it was found to be positively related to psychometric intelligence. Due to the conceptual similarities of TRP and spatial suppression, the present study investigated their mutual interplay in the relation to psychometric intelligence in 273 young adults to better understand the reasons for these relationships. As in previous studies, psychometric intelligence was positively related to a latent variable representing TRP but, in contrast to previous reports, negatively to latent and manifest measures of spatial suppression. In a combined structural equation model, TRP still explained a substantial amount of variance in psychometric intelligence while the negative relation between spatial suppression and intelligence was completely explained by TRP. Thus, our findings confirmed TRP to be a robust predictor of psychometric intelligence but challenged the assumption of spatial suppression as a representation of general information processing efficiency as reflected in psychometric intelligence. Possible reasons for the contradictory findings on the relation between spatial suppression and psychometric intelligence are discussed.

## Introduction

The *temporal resolution power (TRP) hypothesis* explains individual differences in psychometric intelligence by individual differences in the TRP of brain functioning [1]. Within this conceptual framework, TRP is assessed by the timing accuracy and temporal sensitivity in timing tasks such as temporal discrimination, temporal-order judgment, or temporal generalization [2]. Several studies demonstrated that a single latent variable accounted for a substantial portion of common variance in different measures of timing accuracy and temporal sensitivity [2–5]. This latent variable was interpreted as a measure of the brain's fine-tuning capacity

**Funding:** This study was supported by the Schweizerischer Nationalfonds zur Förderung der Wissenschaftlichen Forschung (Grant No. 100014_162377) to THR and SJT. The funders had no role in study design, data collection and analysis, decision to publish, or preparation of the manuscript.

**Competing interests:** The authors have declared that no competing interests exist.

purified from task-specific and error variance [2]. Furthermore, TRP was substantially related to psychometric intelligence, with common variance ranging from 22% [3] to 44% [4]. As an explanation for this relationship, the TRP hypothesis assumes that individuals with higher TRP process information faster and coordinate mental operations better than individuals with lower TRP. Both these factors should contribute to better performance on psychometric intelligence tests [5]. This idea was supported by previous studies. For example, Troche and Rammsayer [6] reported that higher TRP was associated with higher working memory capacity, which in turn led to higher psychometric intelligence. In two other studies, TRP effectively mediated the functional relationship between speed of information processing and intelligence [4, 7]. Hence, higher TRP enables more accurate and, concurrently, faster information processing and, thus, more efficient information processing, which results in better performance on intelligence tests.

Over the last decade, another conceptual framework, referred to as *spatial suppression*, attracted attention due to its possible association with psychometric intelligence [8–10]. On the behavioral level, spatial suppression is evident as a progressively increasing difficulty in perceiving visual motion as stimulus size increases [11]. Spatial suppression is largely restricted to medium and high contrasts, and is particularly strong for briefly presented (e.g., 30 ms) moving grating stimuli [11–13]. This widely replicated result [14] is hypothesized to reflect visual suppression of background motion signals, which in turn promotes rapid segmentation of moving objects [15]. In a typical spatial suppression experiment, participants' task is to correctly identify the direction of the perceived stimulus motion. According to an adaptive algorithm, the presentation time increases after an incorrect response and decreases after a correct response. This results in a motion-direction discrimination threshold (MDD) defined as the shortest stimulus presentation time for which the motion direction could be correctly detected with a given probability [11, 16, 17]. Most interestingly, for high and medium contrast stimuli, the MDD thresholds dramatically increase with increasing stimulus size. In other words, a considerably longer presentation time is needed for larger than for smaller stimuli to correctly identify their motion direction [11]. This increase in MDD thresholds as a function of increasing stimulus size is referred to as spatial suppression. As a commonly used quantification, the spatial suppression index (SI) is computed by subtracting the MDD threshold value for the smallest from the duration threshold for the largest stimulus size used in a spatial suppression task [11, 18–20].

On the neuronal level, spatial suppression has been linked to the function of antagonistic center-surround neurons located in the middle temporal visual area [12, 13, 15, 20–23]. More specifically, the firing rate of these neurons decreases for large high-contrast motion stimuli that, in addition to stimulating the receptive field center, stimulate the antagonistic surrounding region. This results in a diminished neural response to large, high contrast moving stimuli and an overall poorer neural representation of such stimuli [11–13, 20–23].

In a pioneering study, Melnick et al. [10] investigated the correlational relationship between spatial suppression and psychometric intelligence. In two experiments, they obtained substantial correlations of $r = .64$ ($N = 12$) and $r = .71$ ($N = 53$) between SI and psychometric intelligence. Thus, higher intelligence was associated with a more pronounced increase of the MDD threshold from small to large stimuli. Proceeding from these findings, Melnick et al. [10] put forward the idea that spatial suppression reflects the overall neural ability to suppress irrelevant information [14], which is crucial for efficient information processing and, consequently, may explain individual differences in psychometric intelligence [24–27]. More specifically, Melnick et al. [10] concluded that the link between stronger spatial suppression and better performance on intelligence tests indicates that spatial suppression is an index of more efficient information processing via suppression of irrelevant information, not just within visual processing per se but also more broadly.

Although this notion has been supported by two subsequent studies [8, 9], it was at variance with two other ones. The study by Linares et al. [28] investigated the relationship between spatial suppression and intelligence using a between-group design that included a group of schizophrenia patients (N = 33) and a healthy control group (N = 31). The results revealed a link between spatial suppression and a measure of intelligence only in patients with schizophrenia, with no indication for such a relationship for the healthy control group. Furthermore, in a large-scale study with 177 young healthy adults, Troche et al. [29] also failed to confirm a direct functional relationship between spatial suppression and general mental ability.

As a possible explanation of individual differences in intelligence, there is a striking conceptual resemblance between the concepts of TRP and spatial suppression, at least at first glance. First, both are bottom-up approaches as they assume that basic functions of the brain lead to individual differences in higher-cognitive processing and, eventually, in psychometric intelligence. Second, TRP and spatial suppression, in a way, facilitate (or directly reflect) the efficiency of information processing which, in turn, is assumed to be an important aspect of mental ability. Third, both concepts comprise aspects of temporal information processing. While spatial suppression is derived from the time required to correctly identify the direction of perceived stimulus motion, the formation of TRP is based on temporal sensitivity and accuracy.

Despite these similarities, however, several important differences between both concepts become evident at second glance. First, the MDD thresholds heavily depend on (presentation) time required to correctly identify the motion direction of a stimulus, or in other words processing speed is the decisive component of this measure. Spatial suppression, however, is represented by the *difference* between the MDD thresholds for a large and a small stimulus. Therefore, processing speed, which might determine both thresholds, does not necessarily affect the difference between these thresholds. Given the above-outlined relationship between TRP and processing speed, TRP might be related to the thresholds but completely independent of spatial suppression. Second, although spatial suppression can also occur in different modalities, only spatial suppression as a visual phenomenon with its underlying neural mechanisms located in visual brain areas has so far been associated with psychometric intelligence [20, 21]. The TRP-intelligence relationship, on the contrary, does not depend on the modality of a given timing task [3]. More specifically, Haldemann et al. [3] argued that temporal information is processed modality-specific at an initial stage but controlled by a superordinated amodal processing system at a second stage. Most importantly, it was this amodal temporal processing system that was responsible for the relationship between TRP and psychometric intelligence. Thus, while spatial suppression refers to a specific sensory process in the visual system, the scope of TRP is broader and not linked to modality-specific processes.

In view of the above-mentioned similarities and differences between both conceptual frameworks, the functional relationship between TRP and spatial suppression in predicting individual differences in psychometric intelligence remains completely undefined. Therefore, the main goals of the present study were (1) to provide additional evidence for an association between spatial suppression and mental ability and (2) to directly compare the functional relationships between TRP and intelligence as well as between spatial suppression and intelligence, respectively. Another aim of the present study was to systematically investigate the mutual interplay of TRP and spatial suppression in predicting individual differences in intelligence.

For these purposes, a latent variable approach was applied with both TRP and spatial suppression. TRP was represented as a latent variable derived from different timing tasks. For the representation of spatial suppression, we used a similar fixed-links modeling approach as Troche et al. [29]. With this approach, individual differences in the MDD thresholds can be divided into variance systematically increasing with increasing stimulus size and variance

independent of stimulus size. Thus, the latent variable, describing the first kind of variance, can be interpreted as a reflection of genuine spatial suppression. The latent variable, representing the variance not varying with stimulus size, reflects individual differences in the time of stimulus presentation required to correctly detect the motion direction, irrespective of stimulus size [29]. Combining the measurement models of TRP and spatial suppression allowed for the investigation of their functional relationship. In a next step, using latent regression modeling, the relationships between TRP and the *g* factor of psychometric intelligence as well as between spatial suppression and the *g* factor were investigated separately to determine the amount of variance of intelligence shared with TRP and spatial suppression, respectively. Finally, both TRP and spatial suppression were concurrently submitted to the regression model to examine their unique and common variance shared with *g*.

## Methods

### Participants

From an original sample of 296 participants, 23 participants had to be removed due to incorrect test behavior or the results of an outlier analysis. The final sample consisted of 152 women and 121 men ranging in age from 18 to 30 years ($M_{age}$ = 21.6; SD = 2.7 years). All had normal or corrected-to-normal vision and reported no current health issues. Regarding the educational background, 38% of the participants had finished vocational school, whereas 62% had higher educational training. At the time of the study, 47% of the participants were college students, 42% were working in a profession, 10% were still in high school, and 1% were unemployed. For their participation, they received 45 Swiss francs or course credit. All participants signed written informed consent prior to their participation. The study protocol was approved by the local ethics committee of the University of Bern (Faculty of Human Sciences; No. 2016-9-00005).

### Measure of psychometric intelligence

To measure psychometric intelligence, we used a modified short version of the Berlin Intelligence Structure (BIS) test [30] (see also [7, 29]). This version consisted of 18 subtests with six subtests assessing capacity-, six subtests assessing speed-, and six subtests assessing memory-aspects of psychometric intelligence. Each of these six-subtest bundles contained two figural, two numerical, and two verbal subtests. First, the raw scores of the subtests were z standardized before a mean score for capacity, speed, and memory was computed, respectively. Then, by means of a confirmatory factor analysis, the *g* factor was derived from the mean z scores of the three aspects of intelligence.

### Spatial suppression task

The spatial suppression task was designed and used like the one in Melnick et al. [10]. Our goal was to closely match our task to Melnick et al. [10], both in task design and in the experimental equipment. The highest contrast was set to 42%, and the task was presented using a 360 Hz DLP projector (1280 x 720 resolution, 113.7 cd/m² background) as in the study by Melnick et al. [10]. The task was programmed with Matlab [31] to present brief visual grating-like stimuli with a spatial frequency of 1 cycle/˚. These stimuli either moved leftward or rightward on a natively linearized display (178 cd/m² background, 2 cd/m² ambient illumination) with a constant moving speed of 4.8˚/s. Four stimulus sizes were used, subtending a visual angle of 1.8˚, 3.6˚, 5.4˚, and 7.2˚, respectively. The stimulus size was specified by stationary raised cosine spatial envelopes through which moving gratings were shown and, thus, defined as the

visible stimulus diameter (visibility defined as local contrast higher than 1%). The stimulus duration was determined as the full width at half-height of the trapezoidal temporal envelope [20]. To keep the viewing distance constant at 146 cm for each subject, a chin rest was used. Participants gave their responses by using the left and the right arrow keys on a computer keyboard.

At the beginning of the task, participants performed 180 practice trials. Then they completed three blocks with 44 trials per stimulus size, leading to a total of 528 trials and a test duration of about 26 minutes. Within each block, stimulus size varied randomly. Each trial started with a fixation circle, followed by a moving stimulus that was presented in the center of the monitor. Participants then indicated the perceived direction of the drifting grating by pressing the left arrow key when they had perceived a leftward movement and the right arrow key for a perceived rightward movement. They were asked to answer as accurately as possible, with no emphasis on response speed. After their answer, participants received auditory feedback (a 50-ms sine wave tone of 2900 Hz) for a correct answer and no feedback for an incorrect answer. The initial presentation time for each stimulus condition was 80 ms. The presentation time of the next stimulus with the same size was adapted depending on the previous response. In the case of a correct response, presentation time decreased, and after an incorrect response, it increased according to the adaptive Bayesian QUEST-procedure proposed by Watson and Pelli [32]. Based on this procedure, in each block, two estimates of the 82% motion-direction detection threshold were gathered per stimulus size for each participant resulting in six estimates of the threshold for each stimulus size. Because the QUEST procedure requires logarithmic values, the estimated thresholds for motion perception represented the $\log_{10}$ value for presentation time required to produce 82% correct responses, with smaller thresholds indicating better performance. Of the six estimates, the highest and lowest estimates for each stimulus size per individual were discarded, resulting in four remaining thresholds per stimulus size for each individual (see [10]). These four remaining thresholds were then averaged separately for each stimulus size, resulting in one threshold value per stimulus size.

The spatial suppression index (SI) was calculated for each participant by subtracting the $\log_{10}$ of the mean MDD threshold of the smallest stimulus size from the $\log_{10}$ of the mean threshold of the largest stimulus size [19].

## Assessment of temporal resolution power

Temporal resolution power was assessed with the following four timing tasks, which were programmed and presented with E-Prime 2.0 [33].

**Duration discrimination tasks.** In the duration discrimination task with empty auditory intervals (DDE), the intervals were marked by a 3-ms onset and 3-ms offset white noise burst (DDE). These auditory intervals were presented via headphones at an intensity of 70 dB.

The task consisted of 64 trials. Each trial contained a standard interval with a duration of 50 ms and a comparison interval of varying duration. The two intervals were separated by a 900-ms interstimulus interval. In one series of 32 trials, the comparison interval was shorter than the standard interval. In another series of 32 trials, the comparison interval was longer. The two series were interleaved, and the order of standard and comparison interval was randomized and balanced. For each trial, participants' task was to indicate whether the first or the second interval was longer by pressing a designated key on the keyboard. They received visual feedback for 1500 ms on the center of the monitor screen. After an intertrial interval of 600 ms, the next trial started. Following the adaptive weighted-up-down procedure [34], for the first until the sixth trial, the difference between standard and comparison interval of the next trial was increased by 9 ms when the response had been incorrect and decreased by 3 ms when the previous response had been correct. For the seventh until 32nd trial, the respective steps

were 6 ms and 2 ms. With this procedure, the series of 32 trials with the comparison interval being shorter than the standard interval resulted in the 25%-difference threshold (x.25), while the other series resulted in the 75%-difference threshold (x.75). Both thresholds were computed across the last twenty trials of the respective series. As a measure of performance, the difference limen (DL) was calculated by half of the interquartile range [(x.75-x.25)/2], with better performance indicated by smaller values [35].

An additional duration discrimination task (DDF) was used, which had the same procedure as described above for the DDE. However, the stimuli were filled auditory intervals (DDF) of white-noise bursts presented at an intensity of 70 dB. Written instructions and training trials preceded both tasks, which lasted about 7 minutes each.

**Temporal generalization task.**   The temporal generalization task (TG) consisted of 64 trials with a total duration of 5 minutes. The task began with a learning phase in which participants were presented with a standard duration, which was a 75-ms white-noise burst at an intensity of 70 dB presented via headphones. The standard duration was presented five times, and participants were instructed to memorize the duration. Afterward, the experimental phase began, and participants were randomly presented either with the standard duration (75 ms) or with an alternative duration (42 ms, 53 ms, 64 ms, 86 ms, 97 ms, or 108 ms). After each stimulus presentation, they had to decide whether it was the standard stimulus or not by pressing designated keys with "Yes" or "No" on a keyboard. After their response, they received visual feedback in the monitor center for 1500 ms, followed by an intertrial interval of 700 ms. The experimental phase consisted of eight blocks. Within each block, the standard duration (75 ms) was presented twice and each of the six non-standard durations once. The order of the durations was randomized in each block.

As a performance measure, the index of response dispersion was computed by dividing the relative frequency of "Yes" responses to the standard duration by the total of the relative frequencies of "Yes" responses to all seven stimulus durations [36]. A value close to 1 indicates that all the "Yes" answers are closely gathered around the standard duration. For the further analyses, the values of the index of response dispersion were inverted.

**Rhythm perception task.**   The rhythm perception task (RP) consisted of 64 trials. In each trial, a rhythmic pattern of six 3-ms white-noise bursts was presented via headphones at an intensity of 70 dB, leading to five beat-to-beat intervals. Four of these five auditory intervals were held constant with 150 ms, whereas one interval had a variable duration (150 + x) with an initial duration of x = 20 ms. In one series of 32 trials, the third beat-to-beat interval was variable, while in the other series of 32 trials, it was the fourth interval. The two series were presented in interleaved order. For each series, the value of x was adapted according to the weighted-up-down procedure [34]. Thus, depending on the correctness of the previous response, the interval was increased by 4 ms after a correct response and decreased by 12 ms after an incorrect response. After the presentation of the rhythmic pattern, participants had to decide if they had perceived the pattern as regular or irregular by pressing one of two designated keys. Since all patterns had been irregular, participants received no feedback after their response. Instead, the next trial started 700 ms after the response to the preceding trial.

The 75% threshold for the detection of irregularity was calculated for each series and, afterwards, the two thresholds were averaged as a measure of performance. The task lasted about 5 minutes, and written instructions as well as training trials were presented prior to the actual task.

## Time course of the study

The study consisted of two sessions. In the first session, participants completed the psychometric intelligence test (BIS) with a duration of about 90 minutes. In the second session, the

experimental tasks were administered. The second session always started with the spatial suppression task. Afterwards, the timing tasks were administered in a balanced order. Finally, two further experimental tasks were conducted, which are irrelevant for the present purpose. The second session lasted about 120 minutes. Both sessions were separated by about one week.

## Data analysis

The following analyses were conducted with R [37], Version 4.1.0 and R-Studio [38], Version 1.4.1106 using the Hmisc package [39], the rstatix package [40], the ez package [41], the GPArotation package [42] and the lavaan package [43]. Before analyzing the data set, an interindividual outlier detection was computed. For the TRP tasks and for the first threshold of the spatial suppression task (1.8˚), participants were considered outliers and removed from the dataset when they exceeded the mean by three standard deviations. This resulted in a final sample of 273 participants. Then, descriptive statistics were analyzed, followed by correlation analyses and a one-way ANOVA by including the logarithmic thresholds of the spatial suppression task as four levels of a repeated-measures factor. Then, the measurement models for $g$, TRP, and spatial suppression were fit separately, and afterwards, the structural equation models were computed. All models were estimated with robust maximum likelihood estimation. As for fit indices, $\chi^2$ values, comparative fit index (CFI), root mean square error of approximation (RMSEA), and standardized root mean squared residual (SRMR) were determined. If a model fits the data well, then the $\chi^2$ value should be small and not statistically significant [44]. However, the $\chi^2$ statistic is sensitive to sample size, and its $p$ value might be significant, although the predicted model represents the data well [44, 45]. Therefore, we report $\chi^2$(df) but do not consider it in the model evaluation. The other indices were interpreted as good (or acceptable) with the following values [46]: a CFI $\geq$ .95 ($\geq$ .90), RMSEA $\leq$ .05 ($\leq$ .08) and SRMR value $\leq$ .08 ($\leq$ .10).

## Results

### Descriptive statistics

Table 1 shows the descriptive statistics for the four thresholds of the spatial suppression task and for the four timing tasks. In the spatial suppression task, the MDD thresholds increased

**Table 1. Descriptive statistics for the motion-direction detection thresholds in the four conditions of the spatial suppression task, for the difference limina in the duration discrimination tasks (DDE, DDF), for the dispersion index in the temporal generalization task (TG), and the mean 75% difference threshold in the rhythm perception (RP) task in the sample of 273 participants.**

|  | M | SD | Min | Max | Skewness | Kurtosis |
|---|---|---|---|---|---|---|
| **Spatial suppression task** |  |  |  |  |  |  |
| 1.8˚ condition [ms] | 44.69 | 14.22 | 15.71 | 83.63 | 0.25 | -0.48 |
| 3.6˚ condition [ms] | 53.86 | 21.43 | 16.76 | 133.64 | 0.68 | 0.57 |
| 5.4˚ condition [ms] | 72.86 | 34.02 | 13.38 | 277.78 | 1.75 | 6.25 |
| 7.2˚ condition [ms] | 91.45 | 47.32 | 20.90 | 459.23 | 2.75 | 14.39 |
| **TRP tasks** |  |  |  |  |  |  |
| DDE [ms] | 18.33 | 8.22 | 4.95 | 51 | 1.22 | 1.80 |
| DDF [ms] | 8.81 | 3.26 | 3.60 | 23 | 1.67 | 3.82 |
| TG [dispersion index] | 0.66 | 0.12 | 0 | 0.97 | -1.17 | 4.59 |
| RP [ms] | 55.86 | 22.43 | 7.18 | 127.80 | 1.04 | 0.63 |

TRP tasks = Temporal Resolution Power tasks, DDE = Duration Discrimination

with empty intervals, DDF = Duration Discrimination with filled intervals, TG =

Temporal Generalization Task, RP = Rhythm Perception Task. The presented raw values of the spatial suppression task were each multiplied by 2.5 [10].

with increasing stimulus size (also see Fig 1). To investigate if this increase in presentation time with increasing stimulus size was significant, a one-way ANOVA was conducted by including the thresholds of the spatial suppression task as four levels of a repeated-measures factor. Because the Mauchly's test showed a violation of sphericity, the Greenhouse-Geisser correction with $\varepsilon = 0.722$ was applied. ANOVA revealed a significant main effect, $F(2.16, 588.87) = 580.137$, $p < .001$, $\eta^2 = .273$. Bonferroni-corrected pairwise $t$ tests further revealed significant differences between all four thresholds (all $p$s $< .001$). Thus, participants needed a longer presentation time to correctly identify the motion direction when the stimulus became larger. Overall, both the pattern of results and actual thresholds were highly consistent with the results reported in Melnick et al. [10].

The spatial suppression index as the difference between the MDD threshold in the 7.2˚ and the 1.8˚ condition of the spatial suppression task ranged from -.036 to 1.000 (M = .29; SD = .15). Although the spatial suppression effect showed large interindividual differences, it was positive in 99% of the participants.

## Correlational analyses

BIS-Capacity, BIS-Speed, and BIS-Memory correlated positively and significantly with each other ($r_{\text{BIS-Capacity–BIS-Speed}} = .51$, $r_{\text{BIS-Capacity–BIS-Memory}} = .46$, $r_{\text{BIS-Memory–BIS-Speed}} = .39$, all

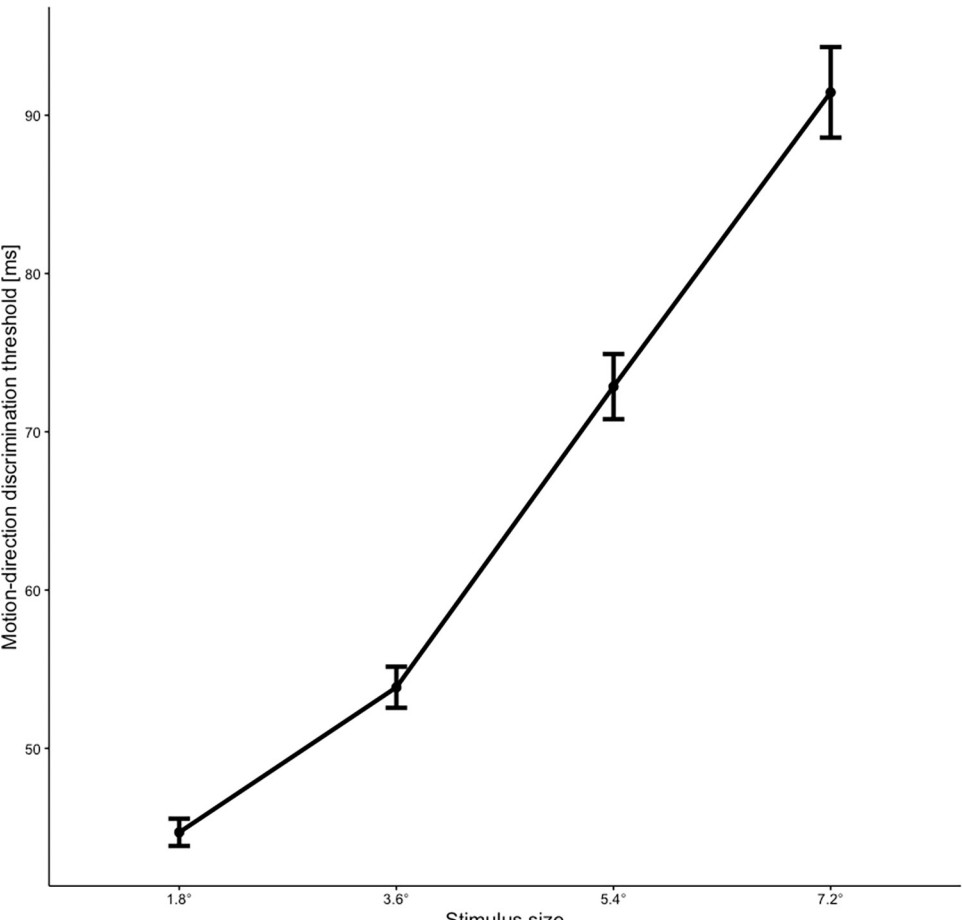

**Fig 1. Line plot of the mean thresholds concerning the four conditions in the spatial suppression task.** The mean per condition (stimulus size, in ms) and its standard errors are presented for 273 participants. For better illustration, the raw values of the presented mean thresholds were each multiplied by 2.5 [10].

**Table 2. Pearson correlations between psychometric intelligence (*g* factor), spatial suppression task, spatial suppression index, and temporal resolution power tasks.**

|  | *g* factor | Spatial suppression task |  |  |  |  | TRP tasks |  |  |  |
|---|---|---|---|---|---|---|---|---|---|---|
| Task |  | 1. | 2. | 3. | 4. | 5. | 6. | 7. | 8. | 9. |
| **Spatial suppression task** |  |  |  |  |  |  |  |  |  |  |
| 1. 1.8˚ | -.17** |  |  |  |  |  |  |  |  |  |
| 2. 3.6˚ | -.24*** | .83*** |  |  |  |  |  |  |  |  |
| 3. 5.4˚ | -.23*** | .69*** | .84*** |  |  |  |  |  |  |  |
| 4. 7.2˚ | -.26*** | .63*** | .76*** | .88*** |  |  |  |  |  |  |
| 5. SI | -.17** | -.16** | .16** | .45*** | .67*** |  |  |  |  |  |
| **TRP tasks** |  |  |  |  |  |  |  |  |  |  |
| 6. DDE | -.24*** | .15* | 18** | .13* | .14* | .04 |  |  |  |  |
| 7. DDF | -.20*** | .18** | .18** | .18** | .20*** | .09 | .36*** |  |  |  |
| 8. TG | -.34*** | .20*** | .20*** | .19** | .19** | .05 | .25*** | .34*** |  |  |
| 9. RP | -.24*** | .14* | .21*** | .22*** | .20*** | .12 | .26*** | .17** | .16** |  |

*N* = 273. TRP tasks = Temporal Resolution Power tasks, DDE = Duration Discrimination with empty intervals, DDF = Duration Discrimination with filled intervals, TG = Temporal Generalization Task, RP = Rhythm Perception Task.

* *p* < .05

** *p* < .01

*** *p* < .001.

*p*s < .001), suggesting a common source of variance. Therefore, a one-factor model was computed by means of a confirmatory factor analysis (CFA). Due to only three manifest variables, the model was exactly identified, resulting in a perfect model fit [44]. McDonald's omega coefficient was ω = 0.72 for the *g* factor extracted from BIS-Capacity, BIS-Speed, and BIS-Memory. The factor scores on this common factor were extracted and interpreted as individual differences in the *g* factor of psychometric intelligence, which were submitted to the following correlational analyses.

As can be taken from Table 2, the MDD thresholds in the four conditions of the spatial suppression task correlated positively with each other. Similarly, positive correlations were also obtained among performance measures in the four timing tasks. Furthermore, performance measures of the four timing tasks correlated significantly with all MDD thresholds of the spatial suppression task.

The *g* factor scores correlated negatively with the four MDD thresholds and the performance measures from the four timing tasks (see Table 2). Since lower thresholds in the timing tasks and the spatial suppression task were indicative of better performance, the negative correlations pointed to positive relationships between psychometric intelligence and performance on the timing tasks as well as the spatial suppression task. Surprisingly, SI was negatively correlated with psychometric intelligence, indicating that higher psychometric intelligence was associated with a smaller SI. This correlational relationship is illustrated as scatterplot in Fig 2. Eventually, in contrast to the MDD thresholds, the SI was not significantly correlated with performance measures from the four timing tasks.

## Measurement models

The TRP factor was derived as a latent variable from the four timing tasks. A one-factor confirmatory factor analysis resulted in a good fit, $\chi^2(2) = 2.770$, $p = .250$, CFI = .990, RMSEA = .038, SRMR = .024. The factor loadings of all four tasks were significant (all *p*s < .001) ranging from

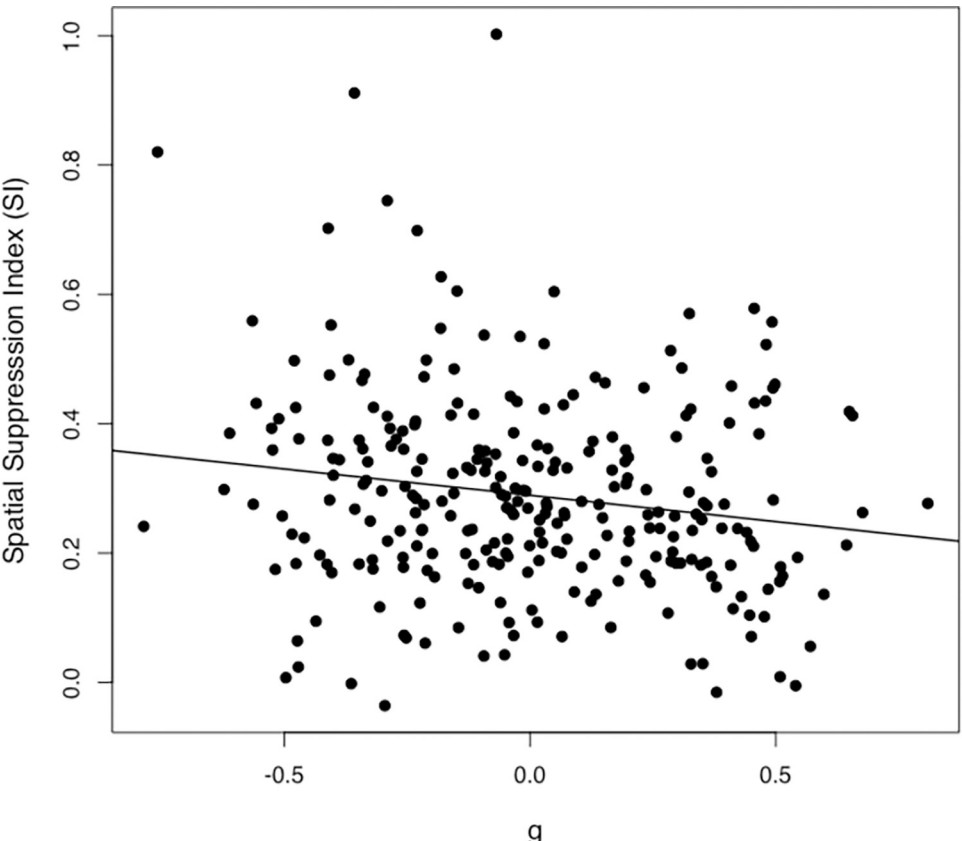

**Fig 2. Scatterplot of the relationship between spatial suppression (spatial suppression index) and *g* in 273 participants.**

.338 for the rhythm perception task to .632 for the duration discrimination task with filled intervals. McDonald's omega coefficient was ω = 0.59 for the TRP factor.

A congeneric model of spatial suppression was first constructed to represent spatial suppression at the latent level with one factor. Confirmatory factor analysis showed a bad model/data fit, $\chi^2(2) = 79.773$, $p < .001$, CFI = .892, RMSEA = .377, except for SRMR = .047. Therefore, a fixed-links modeling approach was used to derive two latent variables from the spatial suppression task (see [29]). The factor loadings of the first latent variable were fixed to the constant value of one for all four thresholds. Therefore, this latent variable is also referred to as "constant latent variable" (SSC in Fig 3). For the second variable, referred to as the "increasing latent variable" (SSI in Fig 3), the factor loadings were set to increase linearly with 0, 1, 2, and 3, respectively, as in the study by Troche et al. [29]. The correlation between SSI and SSC was set to zero. The resulting model was better than the congeneric model but still failed to provide a good description of the data, SBχ²(4) = 41.287, $p < .001$, CFI = .948, RMSEA = .185, SRMR = .074. According to the modification indices (M.I.) provided by the lavaan package, a residual correlation between the second threshold condition (3.6˚) and the third threshold condition (5.4˚) could improve the fit of the model (M.I. = 51.167), suggesting that these two stimulus sizes might have something in common that the latent variables could not explain. When the residual correlation between the second and third threshold condition was allowed, the variances of the first and fourth thresholds showed negative values. Therefore, in the next step, the variances of the first and fourth threshold condition were restricted to values greater than

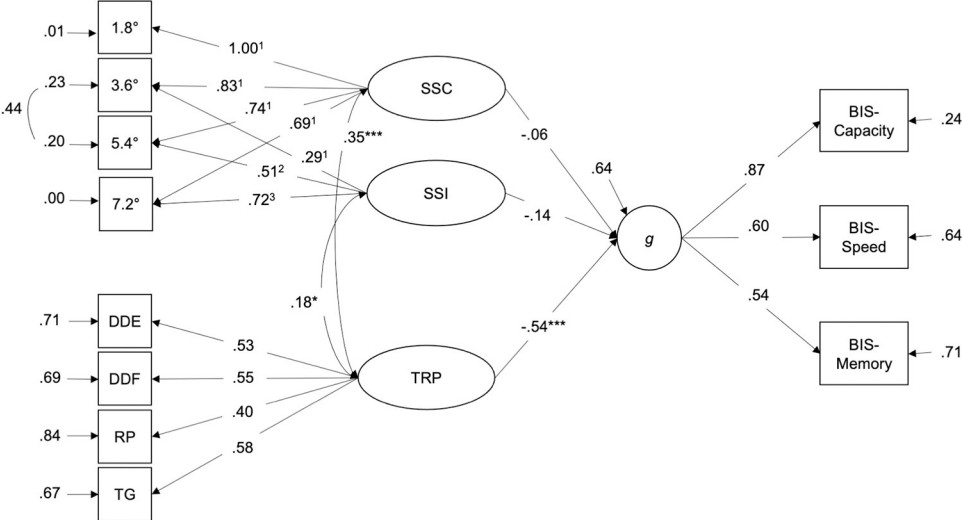

**Fig 3. Final model illustrating the interplay between spatial suppression, TRP, and the *g* factor.** Two latent variables were derived from the spatial suppression task, one representing individual differences in spatial suppression (SSI) and the other representing individual differences in motion-direction discrimination thresholds independent of stimulus size (SSC). Standardized factor loadings and regression coefficients are presented and unstandardized factor loadings for the spatial suppression measurement model are given in superscript. *N* = 273 participants. * p < .05, *** p < .001.

zero. This final model then revealed an acceptable fit according to CFI (= .991) and SRMR (= .066), while the RMSEA with .087 was slightly larger than .080, SB$\chi^2$(3) = 9.162, *p* = .027. The latent variances of both SSC (φ = .022, *z* = 10.345, *p* < .001) and SSI (φ = .003, *z* = 7.771, *p* < .001) were statistically significant. This indicated that both latent variables described a substantial portion of systematic individual differences in the thresholds of the spatial suppression task. When comparing the final model to the congeneric one-factor model, the final model showed a lower AIC (AIC$_{final}$ = -1672.068 compared to AIC$_{congeneric}$ = -1571.579), suggesting that the two-factor solution with the added residual correlation between the second and third threshold condition represented the data better than the one-factor solution. McDonald's omega coefficients were ω = 0.94 for SSC and ω = 0.81 for SSI. To note, SSI represents individual differences in the increase of the MDD thresholds with increasing stimulus size (and, thus, the spatial suppression effect). SSC, on the contrary, reflects general differences in the time required to correctly identify the direction of stimulus movement irrespective of stimulus size.

## Structural equation models

To examine and replicate the relationship between the TRP factor and psychometric intelligence, the measurement models of TRP and *g* were combined, and the *g* factor of psychometric intelligence was regressed on the TRP factor. The model showed an acceptable to good model fit, SB$\chi^2$(13) = 23.904, *p* = .032, CFI = .960, RMSEA = .055, SRMR = .040, and TRP predicted the *g* factor of psychometric intelligence with β = -.572, *p* < .001, thus, explaining 33% of its variance.

The combination of the fixed-links measurement model of spatial suppression and the *g* factor measurement model led to an acceptable to well-fitting model, SB$\chi^2$(13) = 27.146, *p* = .012, CFI = 0.987, RMSEA = 0.063, SRMR = 0.058. Both SSC, β = -.244, *p* < .001, and SSI, β = -.236, *p* = .001, were negatively associated with psychometric intelligence. Thus, participants with higher values in *g* had a less pronounced spatial suppression effect as indicated by the SSI

and, concurrently, generally lower MDD thresholds as indicated by the SSC. Together the two latent variables explained 11% of the variance in *g*.

To investigate the interplay between psychometric intelligence, TRP, and spatial suppression, the relationship between TRP and spatial suppression was examined by combining the above-described measurement models for TRP and spatial suppression and allowing for correlations between TRP and both latent variables extracted from the spatial suppression conditions. The resulting model revealed a good fit, $SB\chi^2(19) = 27.249$, $p = .099$, CFI = .992, RMSEA = .040, SRMR = .050. TRP was significantly and positively correlated with both SSC, $r = .342$, $p < .001$, and SSI from the spatial suppression task, $r = .177$, $p = .022$. Thus, higher TRP was related to faster motion-direction detection (irrespective of stimulus size) and less spatial suppression.

Finally, the prediction of psychometric intelligence by concurrently considering TRP and spatial suppression was examined. The model (Fig 3) showed a good fit, $SB\chi^2(40) = 58.187$, $p = .031$, CFI = .985, RMSEA = .041, SRMR = .050. The TRP factor was still significantly associated with psychometric intelligence, $\beta = -.535$, $p < .001$. However, SSC, $\beta = -.058$, $p = .432$, and SSI from the spatial suppression task, $\beta = -.140$ $p = .057$, did no longer significantly predict psychometric intelligence when TRP was included in the model. Moreover, TRP correlated significantly with SSC, $r = .353$, $p < .001$, and SSI, $r = .180$, $p = .022$. The standardized factor loadings and regression parameters are presented in Fig 3. Thus, the variance SSI and SSC shared with psychometric intelligence could be fully explained by TRP.

## Discussion

The major aim of the present study was to provide further evidence for the functional relationship between psychometric intelligence, on the one hand, and spatial suppression as well as TRP, on the other one. Furthermore, we focused on the mutual interplay of TRP and spatial suppression in explaining variance in psychometric intelligence. The results showed that spatial suppression was negatively related to psychometric intelligence, which contrasts with previous findings of a positive relationship (e.g., [10]). The negative correlational relationship between TRP and psychometric intelligence indicated that higher TRP, and thus higher timing accuracy and temporal sensitivity, was associated with higher psychometric intelligence confirming previous studies (e.g., [7]). Moreover, TRP and spatial suppression were not related at the manifest level but weakly at the latent level, suggesting that they represent widely but not completely independent processes. Higher TRP, however, was moderately related to a shorter time required to correctly identify motion direction irrespective of stimulus size. When psychometric intelligence was regressed on TRP and spatial suppression concurrently, only TRP still explained a significant amount of variance in psychometric intelligence.

Some previous studies reported a positive association between spatial suppression and psychometric intelligence, so that individuals with higher intelligence had a larger spatial suppression effect than individuals with lower intelligence [8–10]. These results suggested that more time was required to identify the correct direction of movement as stimulus size increased and that this was particularly true for individuals with higher (compared to lower) intelligence. In the present study, however, we found a negative relationship between psychometric intelligence and the four MDD thresholds in the spatial suppression task as well as the spatial suppression effect. This result held for the SI as a manifest variable as well as for the SSI latent variable in the measurement model of spatial suppression, which considered the increase of the MDD thresholds with increasing stimulus size across all four conditions of the spatial suppression task.

Procedural reasons for the contradictory findings of a negative relationship between spatial suppression and intelligence in the present study and the positive relationship in previous studies can be largely ruled out. We used the same software and hardware as in the second study by Melnick et al. [10]. The only difference was that our spatial suppression task consisted of four (instead of three) stimulus-size conditions with an additional stimulus size of 5.4˚. The 1.8˚, 3.6˚, and 7.2˚ conditions were also used in the second study by Melnick et al. [10] so that the smallest and the largest stimulus size were identical in the two studies. However, Melnick et al. [10] included a practice session on a separate day, while our participants had a practice session on the day of experimental testing. In unpublished results, Tadin found that practice on a separate day led to less noisy data but had no effect on average thresholds and associated SIs. The fact that both the MDD thresholds as well as their increase with stimulus size were very similar in Melnick et al.'s [10] and the present study corroborated the reliability and replicability of the spatial suppression effect. Its correlation with intelligence, however, seems to be less replicable.

Arranz-Paraíso and Serrano-Pedraza [8] put forward the idea that the lower intelligence level in their sample compared to Melnick et al.'s [10] sample might have led to the weaker (but still positive) relationship between intelligence and spatial suppression in their study. Proceeding from this idea, we reanalyzed our data and submitted only half of the sample with higher psychometric intelligence to the analyses. Even in this subsample with higher intelligence, the spatial suppression index (as well as the four thresholds) still correlated negatively with intelligence, but no longer significantly ($r = -.08$, $p = .382$). Thus, the level of intelligence could not explain the difference between our results and Melnick et al.'s [10] results. Furthermore, when we correlated spatial suppression separately with figural and verbal intelligence subtests, both aspects of intelligence were negatively related to spatial suppression. Thus, the content-related aspects of the intelligence tests in the present and the previous studies were also unlikely to account for the different results. This is also interesting against the background that we used an adapted form of the BIS test as intelligence measure. In contrast, Melnick et al. [10] used the Wechsler Adult Intelligence Scale (WAIS; short form of WAIS-III [47] in Study 1, full version of WAIS-IV [48] in Study 2) and Arranz-Paraíso and Serrano-Pedraza [8] the Reynolds Intellectual Assessment Scales and Screening (RIAS [49, 50]). Thus, at first glance, differences in the way of measuring intelligence cannot be ruled out to account for the divergent relations to spatial suppression. However, as shown by Johnson et al. [51], general intelligence in the sense of a latent $g$ factor shows high stability across different intelligence tests. From this perspective, it seems unlikely that the way of measuring intelligence is a significant reason for the divergent results.

As reviewed in Tadin [14], psychiatric conditions might influence the effect of spatial suppression. A history of major depression, a current schizophrenia diagnosis, or an autism diagnosis have been linked to variations in the strength of spatial suppression. In the present study and the study by Troche et al. [29], however, participants self-reported not to suffer from these psychiatric conditions so that a potential influence of these conditions on the correlation between spatial suppression and psychometric intelligence is unlikely.

A key difference between our study and the two experiments in Melnick et al. [10] was participants' age distribution. This might be important given that several studies showed a negative relationship between age and the spatial-suppression effect [15, 52–54]. While the sample in Melnick et al.'s [10] first experiment had a mean age of 36.0 (± 7.2) years and in the second experiment of 33.1 (± 13.4) years, participants' mean age in the present study was 21.6 (± 2.7) years. When we investigated the influence of age on spatial suppression in the present sample, there was no evidence of any influence of age on spatial suppression, probably because the age range was quite limited with all participants younger than 30 years and 80% of the sample aged

between 18 and 24 years. A similarly young sample was investigated by Troche et al. [29], who also did not observe a positive correlation between spatial suppression and psychometric intelligence. Thus, age-related changes in spatial suppression for participants older than 30 years could not be examined in the present study and the study by Troche et al. [29] so that age might still account for the differences between our results and the results by Melnick et al. [10]. However, Arranz-Paraíso and Serrano-Pedraza [8] as well as Cook et al. [9] also examined participants younger than 30 years and, similar to Melnick et al. [10], observed a positive correlation between psychometric intelligence and spatial suppression. The 31 healthy controls in the study by Linares et al. [28], on the other hand, had a mean age of 38.6 (± 13.8) years and the spatial suppression was unrelated to psychometric intelligence in this sample. Thus, age might be a possible explanation for the inconsistent findings among the relevant studies, but a clear pattern is difficult to discern. It can be stated, however, that the large sample sizes in the present study and the study by Troche et al. [29] strongly argue against a positive link between spatial-suppression strength and psychometric intelligence in adults younger than 30 years.

From a statistical point of view, there are further reasons that could account for the inconsistent results on the relationship between spatial suppression and psychometric intelligence. Regarding the sample size, two studies used small sample sizes (N = 9 in Cook et al. [9]; N = 12 in Melnick et al. [10], Study 1) and two studies medium sample sizes (N = 50 in Arranz-Paraíso and Serrano-Pedraza [8]; N = 53 in Melnick et al. [10], Study 2). All these studies reported a positive correlation between spatial suppression and psychometric intelligence. Linares et al. [28] observed no significant correlation between spatial suppression and intelligence in 31 healthy adults. This latter result was in line with Troche et al. [29], who could not confirm a functional relationship in a sample of 177 participants. The present study with its 273 participants is the only one that even observed a negative (albeit weak) correlational relationship. Thus, the studies, which reported a positive correlation between spatial suppression and psychometric intelligence used rather small samples. This is highly critical given that small sample sizes lead to large confidence intervals around correlations [55]. This problem may be illustrated with Arranz-Paraíso and Serrano-Pedraza's [8] data. In this study, general intelligence correlated significantly neither with the MDD threshold for a small stimulus of 0.7˚, $r = -.213$, $p = .150$, nor with the MDD threshold for a large stimulus of 6˚, $r = .255$, $p = .083$. Given the sample size of N = 47, the 95% confidence intervals ranged from -.47 to +.08 and from -.03 to .51, respectively. Thus, the two correlations did not significantly differ from zero but reached by chance a negative and a positive value, respectively. At this point it is important to note that SI was calculated as the difference between the MDD thresholds for the large stimulus (minuend) and the small stimulus (subtrahend). With this operationalization, the weak (and by chance) positive correlation between the minuend and intelligence was enhanced by the weak (and by chance) negative correlation between the subtrahend and general intelligence. As a consequence, the positive correlation between SI and intelligence reached now statistical significance. This significant correlation, however, seems to be spurious when it is caused by random variation of the correlations between MDD thresholds and general intelligence.

In both the present study and the study by Troche et al. [29] the large sample sizes might have avoided such random variation in the correlation coefficients. Instead, relatively small but significant negative correlations between all four MDD thresholds and intelligence were obtained ranging from $r = -.17$ to $-.26$ in the present study and from $r = -.16$ to $-.19$ in the study by Troche et al. [29]. If we assume that due to the large samples in the latter two studies, the observed correlations between MDD thresholds and intelligence came close to their true values, which did not differ actually, the SI-intelligence correlation was probably less inflated by random variation than in studies with smaller samples. In the present study, the correlations between intelligence and the MDD thresholds for the smallest and for the largest

stimulus did not differ significantly but the latter was more negative than the former one. This small difference was apparently large enough to cause a negative correlation between intelligence and spatial suppression–regardless of whether spatial suppression was measured as SI difference score or as SSI latent variable. Thus, the negative correlation between intelligence and spatial suppression in the present study might be as spurious as the positive correlations reported in other studies and just the result of small and unsubstantial differences between the correlations of intelligence and MDD thresholds for large and for small stimuli. In any case, against the background of the present pattern of results a general *positive* association between spatial suppression and psychometric intelligence seems to be rather unlikely. If and what specific conditions may lead to such a positive association cannot be answered from the present study but need further investigations.

Regarding the functional relationship between TRP and *g*, we were able to replicate previous findings. As in an increasing number of studies (e.g., [3, 4, 7]), TRP explained a substantial portion of variance in psychometric intelligence. More specifically, with 33% explained variance, the communality was only slightly lower than in the studies by Pahud et al. [7] with 36% or Helmbold et al. [4] with 44%. This result underscored the association between psychometric intelligence and higher timing accuracy and temporal sensitivity, as proposed by the TRP hypothesis [5].

The TRP hypothesis by Rammsayer and Brandler [5] proceeded from cognitive models proposing an internal (master) clock underlying the efficiency of information processing and transmission [56–58]. Within the framework of the TRP hypothesis, this higher efficiency is the result of faster and better coordinated information processing. Evidence for the notion that TRP leads to better coordinated mental operations was provided by Troche and Rammsayer [6], who reported that higher TRP led to higher capacity of working memory (WM), which in turn was associated with higher psychometric intelligence. (It should be noted that Jastrzębski et al. [59] reported similar results but argued that WM capacity was the variable, which caused the relation between TRP and psychometric intelligence). Furthermore, the role of TRP as a mechanism underlying the link between speed of information processing and psychometric intelligence was confirmed by several studies [4, 7]. In the present study, TRP was associated with SSC in the measurement model of spatial suppression. In this model, SSC reflected individual differences in speed of information processing as the time required to correctly identify the motion direction (irrespective of the spatial-suppression effect). Individuals with higher TRP needed less time to correctly identify the motion direction than individuals with lower TRP. Furthermore, the relation between SSC and psychometric intelligence decreased substantially when TRP was concurrently considered. Thus, as in previous studies [4, 7], the relationship between speed of information processing (here speed of correctly detecting the motion direction) and intelligence could be explained in terms of TRP. This is particularly interesting because speed of information processing is frequently considered a major source of individual differences in psychometric intelligence [60–62]. A better understanding of the psychophysiological underpinnings of TRP in future studies might help elucidate why TRP so consistently explains the relationship between speed of information processing and psychometric intelligence.

Against our expectation, higher TRP was only weakly associated with a less (instead of a more) pronounced spatial suppression effect as reflected by the SSI latent variable. Several studies supported the idea of a more pronounced spatial-suppression effect being indicative of more efficient information processing [10, 15]. Thus, these two concepts of efficiency seem to be clearly dissociable and only weakly related to each other. This weak relationship, however, was sufficient that TRP explained the complete variance shared between spatial suppression and psychometric intelligence.

In summary, consistent with the TRP hypothesis, the positive relationship between psychometric intelligence and TRP was replicated. TRP was also significantly associated with the portions of variance in MDD thresholds, which were unrelated to stimulus size (i.e., the SSC latent variable) and, thus, reflected speed of information processing. That TRP explained the common variance of this constant latent variable and psychometric intelligence corroborates the assumption that TRP underlies the relation between intelligence and speed of information processing. However, regardless of being operationalized as SI index or SSI latent variable, the spatial-suppression effect was negatively related to psychometric intelligence as well as to TRP. This contradicted the assumption of higher spatial suppression reflecting more efficient information processing [10, 15]. That is, while spatial suppression is a critical mechanism for achieving efficient information processing of visual information, our results called in question prior links between spatial suppression and general brain efficiency as reflected in psychometric intelligence and TRP. If such links exist, it would be important for future research to identify the conditions under which they become effective.

## Author Contributions

**Conceptualization:** Thomas H. Rammsayer, Duje Tadin, Stefan J. Troche.

**Data curation:** Lisa M. Makowski.

**Formal analysis:** Lisa M. Makowski, Stefan J. Troche.

**Funding acquisition:** Thomas H. Rammsayer, Stefan J. Troche.

**Investigation:** Thomas H. Rammsayer, Philipp Thomas.

**Methodology:** Lisa M. Makowski, Thomas H. Rammsayer, Duje Tadin, Philipp Thomas, Stefan J. Troche.

**Project administration:** Thomas H. Rammsayer, Stefan J. Troche.

**Resources:** Thomas H. Rammsayer, Stefan J. Troche.

**Software:** Duje Tadin, Philipp Thomas.

**Supervision:** Thomas H. Rammsayer, Stefan J. Troche.

**Validation:** Lisa M. Makowski, Thomas H. Rammsayer, Duje Tadin, Stefan J. Troche.

**Visualization:** Lisa M. Makowski, Stefan J. Troche.

**Writing – original draft:** Lisa M. Makowski.

**Writing – review & editing:** Thomas H. Rammsayer, Duje Tadin, Stefan J. Troche.

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
