## [Decision Letter · Decision Letter 0]

14 Aug 2022

PONE-D-22-16659On the interplay of temporal resolution power and spatial suppression in their prediction of psychometric intelligencePLOS ONE

Dear Dr. Troche,

Thank you for submitting your manuscript to PLOS ONE. After careful consideration, we feel that it has merit but does not fully meet PLOS ONE’s publication criteria as it currently stands. Therefore, we invite you to submit a revised version of the manuscript that addresses the points raised during the review process.

We look forward to receiving your revised manuscript.

Kind regards,

Nick Fogt

Academic Editor

PLOS ONE

Journal Requirements:

Additional Editor Comments:

Both reviewers are very positive about this manuscript. Please address the following in a revision.

Reviewer #1 indicates that a helpful addition to the manuscript would be a discussion of the state of knowledge on temporal resolution. Please add a brief discussion of these issues.

Reviewer #2 asks for some clarification around the similarities (and/or differences) between the IQ tests used in the current paper and those in other studies. Please address briefly. The reviewer also asks that the authors state whether (or not) there data and the field are trending toward the conclusion that there is no simple relationship between psychometric motion perception and IQ. Please address briefly. Finally, reviewer #2 asks for a discussion of working memory and its potential differential (greater?) impact on TRP tasks versus psychophysical motion tasks. Please address.

Reviewers' comments:

Reviewer's Responses to Questions

**Comments to the Author**

1. Is the manuscript technically sound, and do the data support the conclusions?

Reviewer #1: Yes

Reviewer #2: Yes

2. Has the statistical analysis been performed appropriately and rigorously? 

Reviewer #1: Yes

Reviewer #2: Yes

3. Have the authors made all data underlying the findings in their manuscript fully available?

Reviewer #1: Yes

Reviewer #2: Yes

4. Is the manuscript presented in an intelligible fashion and written in standard English?

Reviewer #1: Yes

Reviewer #2: Yes

5. Review Comments to the Author

Reviewer #1: Review of PLOS ONE #PONE-D-22-16659, “on the interplay of temporal resolution power and spatial suppression in their prediction of psychometric intelligence” by Makowski, Rammsayer, Tadin, Thomas, and Troche.

This paper reports a large (N = 273) psychometric study aimed at clarifying previous inconsistent evidence about correlations of spatial suppression in motion discrimination and temporal resolution with general intelligence. Four of the authors (at U. of Bern, Switzerland) have previously found correlations ( r = ~0. 50) between performance of auditory temporal resolution tasks and general intelligence, and a fifth author (Tadin, at U. of Rochester, U.S.) has co-authored studies that found correlations of spatial suppression in visual temporal resolution in visual motion discrimination with intelligence and other aspects of brain function. A few studies have failed to find the latter correlation, however. The present study is motivated by a conceptual similarity between auditory temporal resolution and spatiotemporal resolution in spatial suppression phenomena in visual motion discrimination, and by the conflicting evidence about correlation of the latter performance with general intelligence.

The present study offers convincing statistical evidence that spatial suppression is not correlated with general intelligence, at least in this population of healthy young adults. The authors used essentially the same visual display methods and software as the experiments that previously found correlations between spatial suppression and intelligence. The present study is similar to a 2018 study by Troche et al., in which Tadin was a co-author, which also found no relationship between spatial suppression and intelligence, but that study did not include the present auditory temporal resolution tasks. The authors note that one possible factor for the conflict with previous findings of a correlation between spatial suppression and intelligence is that the age of the population in this study was younger than in some previous studies. Nevertheless, whether or not age might be relevant factor, the present evidence is quite relevant and useful.

Aside from whether temporal thresholds for motion direction discrimination (MDD), spatial suppression (SI), and the present temporal resolution power (TRP) correlate with general intelligence, numerous basic psychophysical questions remain concerning relations between various measures of temporal resolution and their relevance to various aspects of brain function. The present study focuses on the psychometric rather than psychophysical issues. If revisions are to be made to this manuscript, it would seem useful for the authors to say more about things we do not yet understand about the nature, measurement, and functional implications of temporal resolution. For example, there is relevant literature on the relevance of cognitive models of response time to general intelligence (e.g., research by Anna-Lena Schubert). This is a large general research area that the authors might recognize more explicitly.

I do not have specific required revisions to recommend. The manuscript seems publishable as it stands.

Reviewer #2: - Nicely done contribution to the field of growing evidence between psychometric tasks and general intelligence,

- Super interesting not to replicate some of the original findings but a nice discussion of potential reasons why not. Curious if including a clear reference to the similarity or even identical nature of the IQ tests used here compared to the originals, as there is substantial discussion of the overlap and careful work done in making sure the psychophysics are identical. Often there are standard correlations between IQ tests of different types, and knowing that the IQ tests use here are largely (0.8-1.0) correlated with the IQ tests used in Melnick et al would eliminate another source of variability in these results.

- It seems as if we may be headed towards a conclusion including all of these studies that there is no simple relationship between psychometric motion perception and IQ - this seems perfectly reasonable and I’d be curious to see the authors state if they expect this amid their discussion of some of the statistical issues raised in the discussion.

- There didn’t seem to be much discussion of working memory components during the paper, but it seems clear that many of the TRP tasks involve a heavier component of working memory than some of the psychophysical motion tasks, is there an expectation that that’s part of its greater capture of crucial IQ factors?

6. PLOS authors have the option to publish the peer review history of their article (what does this mean?). If published, this will include your full peer review and any attached files.

Reviewer #1: No

Reviewer #2: **Yes: **Michael Melnick

---

## [Author Response · Author response to Decision Letter 0]

1 Sep 2022

Journal Requirements

Response: We have checked our manuscript regarding PLOS ONE’s style requirements, and we ensure that it meets these criteria. 

Response: We have lifted the embargo so that the repository information is now available by clicking on the stated DOI in the Data Availability Statement.

Response: We have included a full ethics statement in the ‘Methods’ section of our manuscript file by including the ethics committee's full name and information about obtained informed written consent.

Response: We reviewed our reference list and can ensure that it is complete and correct. New references are given in the reference list (concerns reference 54 as well as references 56 to 62). We did not refer to retracted papers.

Responses to reviewer’s comments

Response to Reviewer #1

Reviewer #1: 

This paper reports a large (N = 273) psychometric study aimed at clarifying previous inconsistent evidence about correlations of spatial suppression in motion discrimination and temporal resolution with general intelligence. Four of the authors (at U. of Bern, Switzerland) have previously found correlations ( r = ~0. 50) between performance of auditory temporal resolution tasks and general intelligence, and a fifth author (Tadin, at U. of Rochester, U.S.) has co-authored studies that found correlations of spatial suppression in visual temporal resolution in visual motion discrimination with intelligence and other aspects of brain function. A few studies have failed to find the latter correlation, however. The present study is motivated by a conceptual similarity between auditory temporal resolution and spatiotemporal resolution in spatial suppression phenomena in visual motion discrimination, and by the conflicting evidence about correlation of the latter performance with general intelligence. 

The present study offers convincing statistical evidence that spatial suppression is not correlated with general intelligence, at least in this population of healthy young adults. The authors used essentially the same visual display methods and software as the experiments that previously found correlations between spatial suppression and intelligence. The present study is similar to a 2018 study by Troche et al., in which Tadin was a co-author, which also found no relationship between spatial suppression and intelligence, but that study did not include the present auditory temporal resolution tasks. The authors note that one possible factor for the conflict with previous findings of a correlation between spatial suppression and intelligence is that the age of the population in this study was younger than in some previous studies. Nevertheless, whether or not age might be relevant factor, the present evidence is quite relevant and useful.

Aside from whether temporal thresholds for motion direction discrimination (MDD), spatial suppression (SI), and the present temporal resolution power (TRP) correlate with general intelligence, numerous basic psychophysical questions remain concerning relations between various measures of temporal resolution and their relevance to various aspects of brain function. The present study focuses on the psychometric rather than psychophysical issues. If revisions are to be made to this manuscript, it would seem useful for the authors to say more about things we do not yet understand about the nature, measurement, and functional implications of temporal resolution. For example, there is relevant literature on the relevance of cognitive models of response time to general intelligence (e.g., research by Anna-Lena Schubert). This is a large general research area that the authors might recognize more explicitly.

I do not have specific required revisions to recommend. The manuscript seems publishable as it stands.

Response: We are grateful to Reviewer #1 for his/her positive evaluation of our paper. In the revised version of the manuscript, we give more details about the assumptions of the TRP hypothesis (page 27, line 626 to 629 and page 28, line 645 to 649). We repeat the core assumptions that higher TRP leads to faster information processing and better coordinated mental operations and that these assumptions were supported in previous studies. We also refer to studies, which highlighted the role of information processing as the underlying source of individual differences in psychometric intelligence [1-3]. We also make clear that the psychophysiological basis of TRP is still unclear but promising to learn more about the relation between speed of information processing and psychometric intelligence. 

Responses to Reviewer #2

Reviewer #2: 

Nicely done contribution to the field of growing evidence between psychometric tasks and general intelligence, - Super interesting not to replicate some of the original findings but a nice discussion of potential reasons why not. Curious if including a clear reference to the similarity or even identical nature of the IQ tests used here compared to the originals, as there is substantial discussion of the overlap and careful work done in making sure the psychophysics are identical. Often there are standard correlations between IQ tests of different types and knowing that the IQ tests use here are largely (0.8-1.0) correlated with the IQ tests used in Melnick et al would eliminate another source of variability in these results.

Response: We thank Reviewer #2 for the positive evaluation and his advice to seize up the intelligence measures. We are not aware of a study, which directly investigated the correlational relationship between general intelligence estimates derived from the Wechsler tests and from the Berlin Intelligence Structure test. Süss and Beauducel [4] compared the two measures and pointed to their similarity but did not quantify this. Accordingly, we state in the revised version of the manuscript that the way of measuring intelligence may account for the divergent findings. However, we also refer to the study by Johnson et al. [5]. These authors showed that general intelligence derived from different intelligences tests may differ in their procedure but shows high intercorrelations approaching 1.00 when the test batteries were heterogeneous. Proceeding from this result, it seems unlikely that the way of measuring intelligence is the ultimate reason for the divergent findings. We addressed this on pages 23 and 24, line 533 to 542: “This is also interesting against the background that we used an adapted form of the BIS test as intelligence measure. In contrast, Melnick et al. [10] used the Wechsler Adult Intelligence Scale (WAIS; short form of WAIS-III [50] in Study 1, full version of WAIS-IV [51] in Study 2) and Arranz-Paraíso and Serrano-Pedraza [8] the Reynolds Intellectual Assessment Scales and Screening (RIAS [52,53]). Thus, at first glance, differences in the way of measuring intelligence cannot be ruled out to account for the divergent relations to spatial suppression. However, as shown by Johnson et al. [54], general intelligence in the sense of a latent g factor shows high stability across different intelligence tests. From this perspective, it seems unlikely that the way of measuring intelligence is a significant reason for the divergent results.”

Reviewer #2: 

- It seems as if we may be headed towards a conclusion including all of these studies that there is no simple relationship between psychometric motion perception and IQ - this seems perfectly reasonable, and I’d be curious to see the authors state if they expect this amid their discussion of some of the statistical issues raised in the discussion.

Response: We agree with Reviewer #2 that it is a reasonable conclusion that there is no simple relationship between psychometric intelligence and spatial suppression. To make this clear in the manuscript, we added on page 26, in line 617 to 618 a new sentence (second sentence): “In any case, against the background of the present pattern of results a general positive association between spatial suppression and psychometric intelligence seems to be rather unlikely. If and what specific conditions may lead to such a positive association cannot be answered from the present study but need further investigations.” 

Furthermore, we underscored this by adding a last sentence in the conclusion section (page 28, line 669-670, second sentence): “That is, while spatial suppression is a critical mechanism for achieving efficient information processing of visual information, our results called in question prior links between spatial suppression and general brain efficiency as reflected in psychometric intelligence and TRP. If such links exist, it would be important for future research to identify the conditions under which they become effective.”

Reviewer #2:

- There didn’t seem to be much discussion of working memory components during the paper, but it seems clear that many of the TRP tasks involve a heavier component of working memory than some of the psychophysical motion tasks, is there an expectation that that’s part of its greater capture of crucial IQ factors?

Response: In the revised discussion section, we emphasize now a core assumption of the TRP hypothesis stating that higher TRP leads to better coordinated mental operations, which in turn leads to higher psychometric intelligence (page 27, line 630 to 635). This was supported by Troche and Rammsayer [6], who operationalized “coordination of mental operations” as WM capacity. Thus, when Reviewer #2 asks for expectations, this is now included. It should be noted, however, that Jastrzebski et al. [7] also investigated this interplay between TRP, WM capacity, and psychometric intelligence but concluded that WM capacity is the ultimate variable, which causes the relationship between TRP and intelligence. We refer to this study and its alternative view. We would like to refrain from a longer discussion of this point because we do not think that it is WM capacity what is reflected by TRP (given, for example, that primarily simple decisions are required by the tasks and demands on memory are very low). A final decision cannot be made at the present point of knowledge so that we could just discuss in length the arguments, which are irrelevant for the present purpose.

References 

1. Jensen AR. Clocking the mind: Mental chronometry and individual differences. 1st ed. Amsterdam, Boston, London: Elsevier; 2006.

2. Schubert A-L, Frischkorn GT. Neurocognitive psychometric of intelligence: How measurement advancements unveiled the role of mental speed in intelligence differences. Current Directions in Psychological Science. 2020;29(2): 140-146. doi:10.1177/09637214989635

3. Sheppard LD, Vernon PA. Intelligence and speed of information-processing: A review of 50 years of research. Personality and Individual Differences. 2008;44(3): 535-551. doi:10.1016/j.paid.2007.09.015

4. Süss H-M, Beauducel A. Faceted models of intelligence. In: Wilhelm O, Engle RW. Handbook of understanding and measuring intelligence. Sage Publications; 2005. pp. 313-322. doi:10.4135/9781452233529

5. Johnson W, Nijenhuis J te, Bouchard TJ. Still just 1 g: Consistent results from five test batteries. Intelligence. 2008;36: 81–95. doi:10.1016/j.intell.2007.06.001

6. Troche SJ, Rammsayer TH. The influence of temporal resolution power and working memory capacity on psychometric intelligence. Intelligence. 2009;37: 479–486. doi:10.1016/j.intell.2009.06.001

7. Jastrzebski J, Kroczek B, Chuderski A. Galton and Spearman revisited: Can single general discrimination ability drive performance on diverse sensorimotor tasks and explain intelligence? Journal of Experimental Psychology: General. 2021;150(7): 1279-1302. doi:10.1037/xge0001005

---

## [Editor Report · Decision Letter 1]

5 Sep 2022

On the interplay of temporal resolution power and spatial suppression in their prediction of psychometric intelligence

PONE-D-22-16659R1

Dear Dr. Troche,

We’re pleased to inform you that your manuscript has been judged scientifically suitable for publication and will be formally accepted for publication once it meets all outstanding technical requirements.

Kind regards,

Nick Fogt

Academic Editor

PLOS ONE

Additional Editor Comments (optional):

The authors have thoroughly addressed the reviewer comments.
---

## [Editor Report · Acceptance letter]

11 Sep 2022

PONE-D-22-16659R1 

On the interplay of temporal resolution power and spatial suppression in their prediction of psychometric intelligence 

Dear Dr. Troche:

I'm pleased to inform you that your manuscript has been deemed suitable for publication in PLOS ONE. Congratulations! Your manuscript is now with our production department. 

Kind regards, 

on behalf of

Dr. Nick Fogt 

Academic Editor

PLOS ONE